# The Effect of Brush Plate Structure and Operating Parameters on the Energy Consumption of Electrolytic Cells

Shengxian Yi, Zhongjiong Yang *, Liqiang Zhou and Gaofeng Zhang

State Key Laboratory of High-Performance Complex Manufacturing, School of Mechanical and Electrical Engineering, Central South University, Changsha 410083, China; shengxian21@126.com (S.Y.); csurobert@csu.edu.cn (L.Z.); zhanggao2021@126.com (G.Z.)
* Correspondence: yzj7072@126.com

**Abstract:** The nickel powder brush plate is a core component of the direct contact between the cleaning machine and cathode plate of an electrolyzer, and its movement in the electrolytic cell will affect the energy consumption of the electrolyzer. In order to optimize the structure of the brush plate, a cleaning trolley brush plate was taken as the research object, a mathematical model of its electrolyzer was established, and the reliability was subsequently verified. The influence of the structural and operating parameters of the brush plate on the energy consumption of the electrolytic cell was studied. The research results show that additional energy consumption is the lowest in the process of cleaning a return grooved brush plate. Brush plates with a large slotting area have less impact on the energy consumption of the electrolyzer. The slotting method, where the anodes are arranged directly opposite each other and relatively concentrated, can be adapted to render a more uniform current density distribution on the anode surface, with lower energy consumption and less variation in voltage and current. With the increasing number of slots from one to three, the current density distribution on the anode surface became more uniform, with a reduction in the variation range of the slot voltage and current in the branch where the cathode plate was cleaned and a decreased energy consumption. With the linear increase in brush cleaning speed, the impact time of the brush plate on the electrolyzer decreased nonlinearly, and as the extent of this decrease gradually diminished, the additional energy consumption showed the same trend. These research results were then used as a basis for optimizing the existing commonly used empirical C-brush plates. Following optimization, the current density distribution on the anode surface was found to be more uniform, the variation amplitude of tank voltage was reduced by 34%, the current drop amplitude of the branch circuit where the brushed cathode plate was located was reduced by 39%, the impact time on the current field of the electrolytic tank was reduced by 40%, and the additional energy consumption was reduced by 50.9%. These results can be served as a reference for further theoretical research related to brush plates.

**Keywords:** clean brush nickel powder; brush plate; structural parameters; cleaning speed; energy consumption

## 1. Introduction

Nickel, a heavy metal, is extremely harmful to the environment and the human body. Therefore, safety in the treatment of nickel in industrial production has always been an important issue. In the interest of health and environmental protection, there is consensus from industry regarding the need for technology and equipment to replace the traditional manual handling of nickel [1].

Currently, there are many techniques for nickel removal, such as vibration, pulsed power, electrochemistry, and adsorption, and numerous studies have been conducted in related fields have also conducted research. For instance, Cordero et al. presented a theoretical and experimental evaluation, comparing fluorescence background removal

approaches for Raman spectra and the advantages and disadvantages of the fluorescent background removal method [2]. Feng et al. used hyperspectral imaging technology to identify and remove transgenic maize seeds [3]. Ghasemi et al. used pulsed power to degrade various pollutants and studied the effect of pulsed power treatment time on the degradation rate of different pollutants [4]. Wai Siong Chai and Pau Loke Show et al. investigated the performance of novel materials in removing heavy metals from wastewater, which contributed to the removal of metal ions in water purification and wastewater treatment [5,6].

However, most nickel is removed electrochemically during metallurgical production. Wang et al. removed low concentrations of nickel ions in electroplating wastewater via a combination of electrodialysis and electrodeposition [7]. Saied et al. evaluated the performance of novel absorbents used for the removal of nickel ions from aqueous solutions; the results showed that solution-cast sulfated chitosan and polyvinyl alcohol membrane system had high adsorption removal efficiency [8]. Yu et al. found that nickel ions in electroplating wastewater can be removed using a biomineralization method, where a nickel ion removal efficiency of 76.41% could be reached with the optimum content of organophosphate monoester [9]. Ng et al. investigated the performance of a rotating packed bed (RPB) contactor for nickel removal from water in terms of operating parameters and mode. They showed that the rotational speed and mode of operation had significant effects on nickel adsorption performance [10]. Du et al. prepared ion-imprinted composite membranes for the selective electrochemical removal of heavy metal ions from wastewater. It was found that nickel ions could be removed from the membranes by applying a strong potential [11]. Pătescu et al. used hydroxyapatite composites to remove nickel from multi-metal aqueous systems, and their results showed that these composites could be used as effective adsorbents of heavy metals from aqueous synthetic solutions and wastewater [12]. Caprarescu et al. used copolymers to remove nickel ions from synthetic wastewater, which showed that the active ingredients of the natural extracts could provide more active sites and effectively remove nickel ions from synthetic wastewater and improve the thermal stability of membranes [13]. Koene et al. investigated different electrochemical methods for removing nickel, and their study showed that a minimum concentration of about 1 mol m$^{-3}$ was possible following direct electrolysis and normal electrodialysis, and approximately $10^{-2}$ mol m$^{-3}$ was achieved following ion exchange-assisted electrodialysis [14].

Generally, scholars have focused on chemical properties and reaction mechanisms when studying nickel removal, while surprisingly little attention has been devoted to the structure of the removal equipment. If the cathode plate—an essential component of automatic cleaning equipment involved in the electrolytic production of nickel powder—comes into contact with the electrolyte, it will cause the current and voltage in the electrolyzer to change, and it can be seen from the Equation ($W = \frac{U^2}{R}T$) that when the voltage changes, it causes the energy consumption. Additionally, the machine's structural and operating parameters have a significant influence on electrolytic reactions and the quality of the electrolyzer in terms of cleaning nickel powder. However, most existing structural forms are empirical designs developed by workers and not based on any systematic and deep theoretical research on the structural forms of brush plates.

In this paper, the brush plate of a brush cleaning trolley was the object of study. From the perspective of electrochemical theory, a mathematical model of an electrolytic cell under the brush plate operating conditions was established based on the current density distribution theory. The effects of brush plate structural and operating parameters on the energy consumption of the electrolyzer were studied when brushing nickel powder. Simultaneously, the design of existing commonly used empirical C-brush plates was optimized, and more reasonable structural and operating parameters for the brush plate were obtained and verified by comparison. The additional energy consumption of the optimized brush plate is 50.9% lower than that of the commonly used empirical C-brush plate. This study could serve as a theoretical reference for further research into reducing

the additional energy consumption of electrolytic cells and for the structural design and research of brush cleaning machines.

## 2. Model Building

### 2.1. Numerical Model

A schematic diagram of the working principle of a nickel powder brush cleaner is shown in Figure 1. The electrolytic cell consists of a cathode plate, an anode, and an electrolyte that fills the space between the cathode and the anode and is constantly replenished. The electrolyte is chloride. The brush wire area of the brush plate is in contact with the cathode plate.

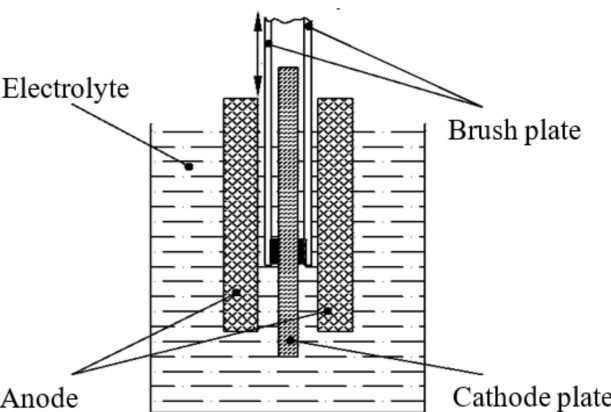

**Figure 1.** Schematic diagram of the working principle of a nickel powder cleaning brush.

The movement of the two brush plates of the cleaning machine is simultaneously up and down along the gap between the cathode plate and the anode. Through the friction between the brush wire area and the two sides of the cathode plate, the nickel powder adhering to the surface of the cathode plate is dislodged and settled at the bottom of the electrolyzer. According to current density distribution theory [15], the net current density of the electrolyte is expressed as the sum of the fluxes of all ions:

$$J_l = F\sum_i z_i N_i \text{ and } \nabla \cdot J_l = 0 \tag{1}$$

where $J_l$ is the current density vector of the electrolyte (A/m$^2$), $F$ is Faraday's constant (C/mol), $N_i$ is the flux of substance $i$ (mol/(m$^2$·s)), and $z_i$ is ionic charge number.

The ionic flux of the electrolyte can be described by the Nernst–Planck equation [16]:

$$N_i = -D_i \nabla c_i - z_i u_{m,i} F c_i \nabla \phi_l + c_i u \tag{2}$$

The Equation describes the flux of solute substances in the electrolyte using diffusion, migration, and convection. Among them $c_i$ is the concentration of ion $i$ (mol/m$^3$), $D_i$ is the diffusion coefficient (m$^2$/s), $u_{m,i}$ is the migration rate (s · mol/kg), $\phi_l$ is the electrolyte potential (V), and $u$ is the velocity vector (m/s).

In electrolytic production, the electrolyte solution is constantly being replenished, monitored, and adjusted. It can be assumed that there is no concentration gradient in the electrolyte. Therefore, the diffusion term in Equation (2) can be ignored. Electrochemical reactions at the electrode–electrolyte interface proceed rapidly. Since the polarization resistance caused by electrode polarization is small compared to electrolyte resistance [17], the influence of electrode dynamics can be ignored. Therefore, it can be assumed that the potential difference at the electrode–electrolyte interface does not deviate from the

equilibrium value, assuming that the electrolyte is electrically neutral. Thus, the convection term can be disregarded. The substitution of Equation (2) into Equation (1) results in

$$J_l = -F^2 \nabla \phi_l \sum_i z_i^2 u_{m,i} c_i \text{ and } \nabla \cdot J_l = 0 \tag{3}$$

Next, replacing $z_i$, $u_{m,i}$, $c_i$, and the Faraday constant with the conductivity $\kappa$ ($1/(W \cdot m^2)$) equivalent gives

$$\nabla \cdot (-\kappa \nabla \phi_l) = 0, \tag{4}$$

i.e., the current density vector in the electrolyte:

$$J_l = -\kappa \nabla \phi_l \tag{5}$$

where the value of electrolyte conductivity is in the range of 14.4–14.8 S/m. For convenience, the intermediate value of conductivity is taken as $\kappa$ = 14.6 S/m. [18]

The relationship between voltage and current in an electrolytic cell can be described using the current density distribution theory [19]. In order to more accurately describe the current density distribution in the electrolyzer, both the electrolyte ohmic and polarization resistance at the electrode–electrolyte interface were considered. The Butler–Volmer equation was used to describe the current density distribution pattern in the electrolyte [20–22].

Current is not uniformly distributed on an electrode surface, as it is related to the different resistance values encountered when the current passes through the electrolyte between the cathode and the anode. As such, it is correlated with geometric factors [23]. Thus, the anode current density when the brush plate is running in the electrolyzer is as follows [15]:

$$I_a = J_0 \frac{(3S_0 + S(t))}{4S_0} \left( \frac{(\alpha_a + \alpha_c)F}{RT} \right) \eta \tag{6}$$

where $J_0$ is exchange current density ($A/m^2$), $R$ is gas constant ($J/(K \cdot mol)$), $T$ is the thermodynamic temperature (K), $\alpha_a$ and $\alpha_c$ comprise the cathodic symmetry factor, $\eta$ is superpotential (V) which are the same for the cathode and anode, $S_0$ is the normal flux area ($m^2$) between the cathode and anode when there is no brush plate, and the normal flux area is the area of the orthogonal projection surface between the cathode and the anode that is immersed in the electrolyte, $S(t)$ is the normal flux area ($m^2$) between the cathode and anode when the brush plate enters the electrolyte at a depth of $vt$ (m), $v$ (m/s) is the speed of the brush plate when clearing the nickel powder, and $t$ is time (s).

The superpotential of the cathode and anode is the difference value between the electrode potential and the equilibrium potential of the electrode reaction [24]

$$\eta_c = -\phi_{l,c} - E_{eq,c} \eta_a = E_{slot} - \phi_{l,a} - E_{eq,a} \tag{7}$$

where $\phi_{l,c}$ and $\phi_{l,a}$ are the electrolyte potentials adjacent to the cathode and anode, respectively, $E_{slot}$ is the tank voltage, $E_{eq,a}$ is the anode equilibrium potential, and $E_{eq,c}$ is the cathode equilibrium potential.

$S(t)$ is related to the structure of the brush plate, the size, and the arrangement of the cathodes. Different structural forms of brush plates have different appearances, but the principle remains the same. An existing C-shaped brush plate is shown in Figure 2, where H is the maximum depth the brush plate can enter the electrolyte, d is the width of the upper part of the brush plate, h is the height of the brush filament area, and r is the half-width of the return slot.

The structure and relative position of the cathode and anode are shown in Figure 3, where $h_a$ is the depth of the anode immersed in the electrolyte, $b_a$ is the width of the anode, $b_1$ is the width of the gap of adjacent anodes, $h_c$ is the depth of the cathode plate immersed in the electrolyte, and $b_c$ is the width of the cathode plate. Moreover, the anodes are evenly distributed based on the center line of the cathode plate.

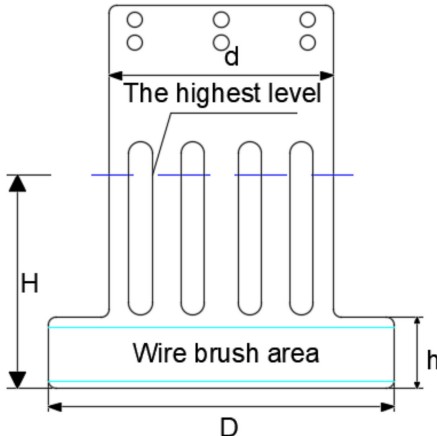

**Figure 2.** C-brush plate.

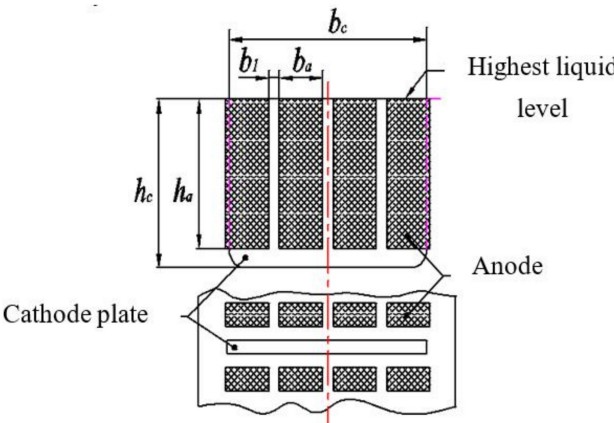

**Figure 3.** Structure and relative position of cathode and anode.

When the bottom edge of the brush plate runs to the standard electrolyte liquid level depth of 0.2 m, the normal flux surface between the cathode and anode of the group is shown in the shaded part of Figure 4.

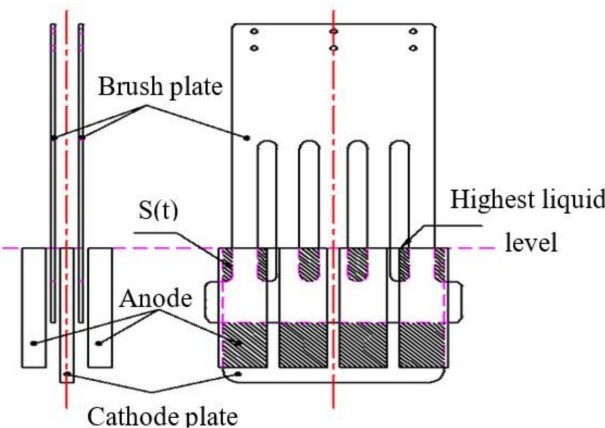

**Figure 4.** Schematic diagram of normal flow surface between cathode and anode.

As can be seen from Figure 4, $S(t)$ is associated with the structure and operating parameters of the brush plate. Thus, for the C-brush plate in Figure 2 $S(t)$ can be expressed as a segmentation function as follows:

①    If $0 \leq vt \leq h$:

$$S(t) = S_0 - 2(b_c - 3b_1)vt \tag{8}$$

②    If $h < vt \leq h + r$:

$$S(t) = (h_a - vt)(2b_a - 3b_1 + d) + 8\left[\arccos\left(\tfrac{r+h-vt}{r}\right)\cdot r^2 - \sqrt{(2r + h - vt)(vt - h)}(r + h - vt)\right] \tag{9}$$

③    If $h + r < vt \leq h_a$:

$$S(t) = 2[(8r - b_c + 3b_1)vt - 8r(h + r) + h_a(b_c - 3b_1) + 2\pi r^2] \tag{10}$$

④    If $h_a < vt \leq H$:

$$S(t) = 4\left[4rvt - 4r(h + r) + \pi r^2\right] \tag{11}$$

When the brush plate enters the electrolyte to clear the nickel powder, the total current $I_a$ drops differently because of the different mechanisms of the brush plate. The magnitude of the fluctuation is mainly related to the change in the normal flux area between the cathode and anode.

During the normal electrolysis process, the total current $I_z$ of the electrolyzer essentially remains constant. It is assumed that the total current of the electrolytic cell decreases uniformly during the entry of the brush plate into the electrolyte. The total current of the electrolyzer drops to the maximum value when the brush plate enters the limit depth. Given the same structural form of the brush plate, as the brush plate enters the electrolyte, the total current in the electrolytic cell varies uniformly with the depth of brush plate entry. As such, one can obtain:

$$I_a = I_z + \frac{I(S_B)vt}{H} \tag{12}$$

Thus, substituting Equation (12) into Equation (6) gives

$$I_z = J_0 \frac{(3S_0 + S(t))}{4S_0}\left(\frac{(\alpha_a + \alpha_c)F}{RT}\right)\eta - \frac{I(S_B)vt}{H} \tag{13}$$

From Equation (15) and the relationship between tank voltage and overpotential, the changes in electrolyzer tank voltage with time in the nickel powder cleaning process can be derived. Additionally, by integrating the current density of the anode surface of the branch where the cathode plate is cleaned, the changes in the current flowing in the branch where the cathode plate is located can be obtained.

### 2.2. Simulation Model

As shown in Figure 5, our electrolyzer model was built by using COMSOL. The dark part is the simplified model of the anode of the electrolytic cell, where each branch has eight electrolyzers. The flat recess is a simplified model of the electrolyzer cathode plate, and there are four brush plates in one electrolyzer. The solid part between the cathode and the anode is the electrolyte. Using the standard depth of the brush plate immersed in an electrolyte with 0.2 m as an example, a model containing the brush plate is shown in Figure 5b, where the dark part is the surface of the brush plate. Since the brush plate was not involved in the electrochemical reaction in the electrolyte, it could be replaced by immersing its surface in the electrolyte.

In the physics field of COMSOL, we selected "secondary current distribution" in the electrochemical module. The electrolyte conductivity was 14.6 S/m. The initial value of the electrolyte potential was set to phil = $\left[(E_{slot} - E_{eq-a} - E_{eq-c})/2\right]$ V. The initial value of the potential was set phis = 0 V. Cathode and anode spacing were fixed at 30 mm, and the cathode boundary was set as shown in Table 1.

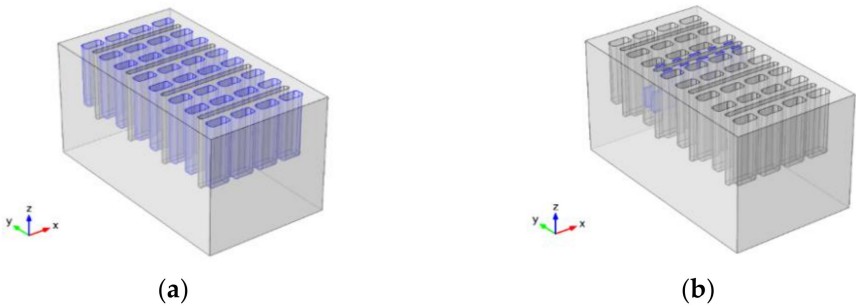

**Figure 5.** Electrolysis tank model: (**a**) without brush plate; (**b**) with brush plate.

**Table 1.** Cathode and anode boundary.

| Electrolytic Tank Voltage | Reaction Temperature | Exchange Current Density of Cathode and Anode | Cathodic Symmetry Factor | Anodic Equilibrium Potential | Cathodic Equilibrium Potential |
|---|---|---|---|---|---|
| 12.6 V | 368 K | 300 /m$^2$ | 0.5 | 0.24 V | 0 V |

Boundary layer grids were set on the surfaces of the cathode and anode to increase the grid cell density. The electrode dynamics expression was set to "Linear Butler Volmer". The anode surface was coupled with the "integration" and "averaging" operations. By calculating the ratio of the absolute and average values of the anode surface current density, the dimensionless current density distribution on the anode surface could be obtained to analyze the influencing factors.

## 3. Validation of the Model

### 3.1. Experiment

In order to verify the accuracy of the simulation and exclude the possible influence of data variability due to different experimental objects, experiments were conducted on different brush plates, and the results were compared with those from the simulation. The electrolyte was still chloride. The main experimental equipment and experimental schematic are shown in Figures 6 and 7.

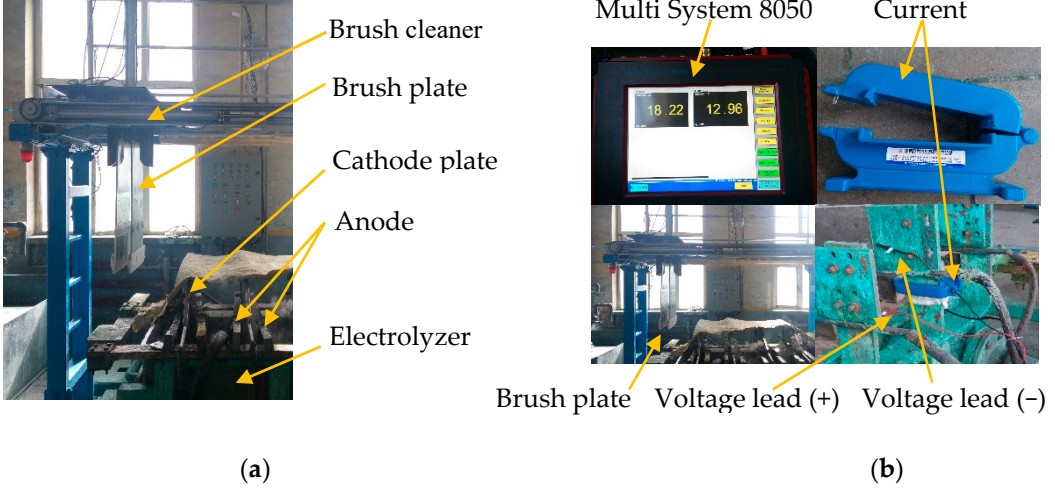

**Figure 6.** *Cont.*

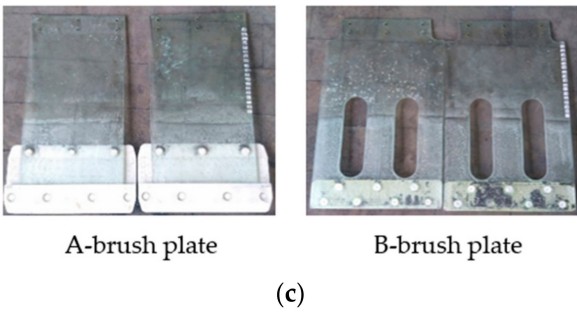

A-brush plate                 B-brush plate

(**c**)

**Figure 6.** (**a**) Main experimental equipment. (**b**) Main experimental equipment. (**c**) Main experimental equipment.

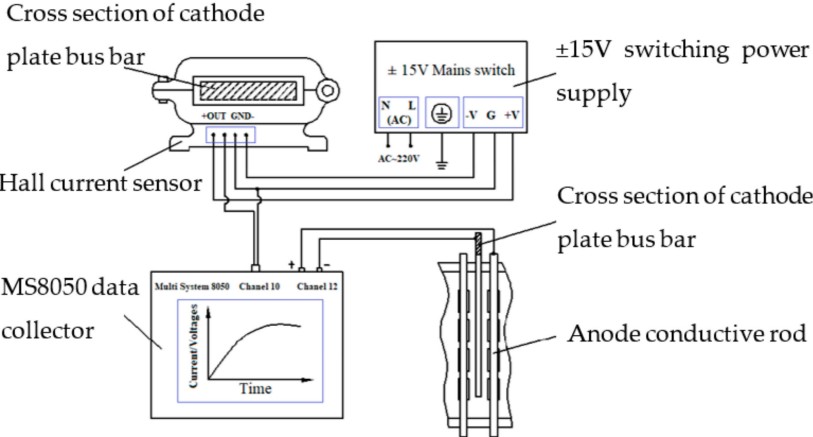

**Figure 7.** Experimental schematic.

Figure 6a was the experimental platform of the new brushing machine. The cathode plate was fixed on the cathode bus bar behind the electrolytic cell by bolts, and the anode was hung on the anode conductive rod. The electrolytic cell was located directly under the cleaning machine. There were four cathode plates in the same electrolytic cell, and the four branches were connected in parallel.

As shown in Figure 6b, the MultiSystem 8050 multifunctional data collector simultaneously recorded and showed the collected tank voltage and current signals at all times. The current sensor converted the strong current signal in the connected cathode plate into a weak current signal and transmitted it to the MultiSystem 8050 multifunctional data collector. The voltage signal between the cathode and anode was transmitted to the channel 12 of the data collector by two voltage leads and recorded in real-time.

As shown in Figure 6c, two brush plates were made of PEEK material. Brush plate A and brush plate B had a different number of grooves and structural shapes, and the accuracy of the simulation data was verified by comparison.

A brush plate was mounted on the cleaner, and the operating mode of the cleaner was switched to "Manual". The operating speed of the cylinder rod of the brush cleaner was adjusted in the range of 75–300 mm/s. As shown in Figure 7, ±15 V switching power supply is used to supply power to Hall current sensor, the Hall current sensor converted the detected high current into 0–20 mA (1 mA represents the actual 100 A), which was transferred to channel 10 of the MultiSystem 8050 data collector, and the voltage and current signals were recorded and saved in real-time. Simultaneously, the voltage leads of the anode conductive rod and cathode plate were connected, and the tank voltage was transmitted to channel 12 of the data collector and recorded in real-time. Considering that the magnetic field generated near the electrolyzer could interfere with the measured electrical signals, the measured signals were transmitted using magnetically shielded wires with a strong antimagnetic interference ability.

### 3.2. Validation of the Model

The tank voltage and the current variation of the brushed branch circuit following the nickel powder brushing process are shown for the two above-mentioned types of brush plate electrolytic cells in Figures 8 and 9. Typically, the curve has a roughly symmetrical characteristic. Regardless of whether the brush plate moved downward or upward, the speed and response time of the brush plate corresponding to the position at the same distance from the bottom end are the same, which is related to the symmetry of the brush plate itself in a working cycle.

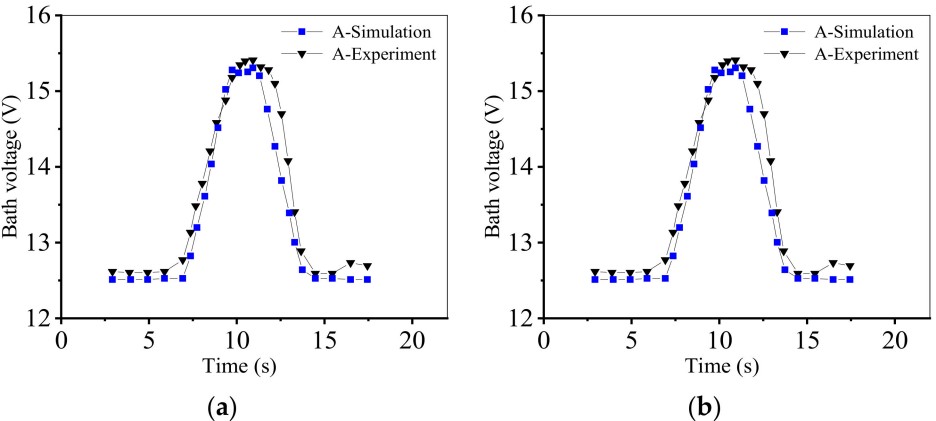

**Figure 8.** Law of the brush plate slot voltage variation with time. (**a**) A-brush plate; (**b**) B-brush plate.

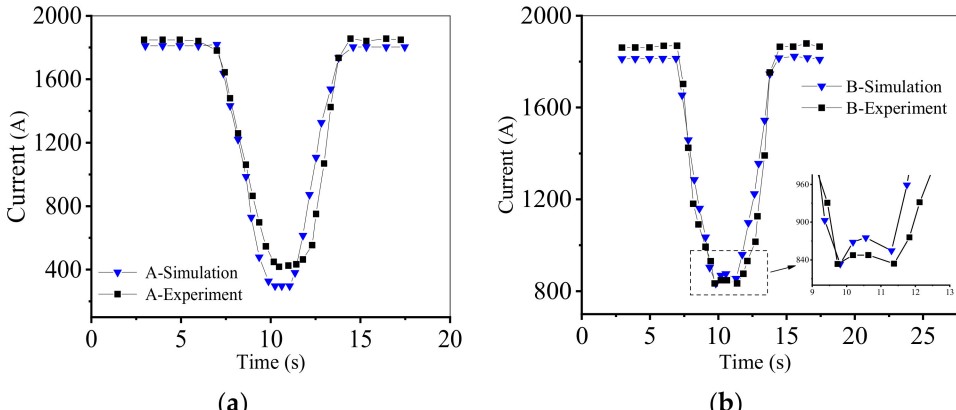

**Figure 9.** The change law of the branch current of the brush plate vs. time during cleaning. (**a**) A-brush plate. (**b**) B-brush plate.

The simulated values of the model and the real values of the experiment could be quantified by the mean values and maximum values of the relative errors [25–27].

$$AARE(\frac{0}{0}) = \frac{1}{N}\sum_{i=1}^{N}\left|\frac{E_i - S_i}{E_i}\right| \times 100\frac{0}{0} \tag{14}$$

$$\varepsilon_{\max} = \max\{\varepsilon_1, \varepsilon_2, \varepsilon_3 \cdots \varepsilon_i\} \tag{15}$$

In $\varepsilon_i = \left|\frac{E_i - S_i}{E_i}\right| \times 100\frac{0}{0}$, $E$ indicates the experimental measurement value, and $S$ denotes the model simulation value.

From Figure 8, it can be seen that the average value of the relative error of the A-brush plate between the simulation and experimental results was 5.6%, and the maximum value of the relative error was 10.9%. The average value of the relative error of the B-brush plate was 5.4%, and the maximum value of the relative error was 8.4%.

From Figure 9, it can be seen that the average value of the relative errors of the A-brush plate current between the simulation and experimental results was 7.3%, and the maximum value of the relative error was 21.4%. The average value of the relative error of the B-brush plate was 7.4%, and the maximum value of the relative error was 17.9%.

Thus, although there were individual cases of significant relative errors, the simulation results of the slot voltage and current of the two types of brush plates were in high agreement with the experimental results, indicating the errors were within an acceptable range. Therefore, the simulation results of the numerical model in this paper can adequately describe the changes in the slot voltage and current of a brushed branch.

## 4. Simulation Analysis

### 4.1. The Influence of Structural Parameters on the Energy Consumption of Electrolyzer

4.1.1. Slot Shape

In order to facilitate comparison and analysis, three types of brush plates—circular groove, diamond groove, and elliptical groove—were designed; they are shown in Figure 10A–C, respectively. For each type, the total area of the grooves of a single brush plate was designed to be 60,000 mm$^2$, and the area of a single groove was designed to be 15,000 mm$^2$.

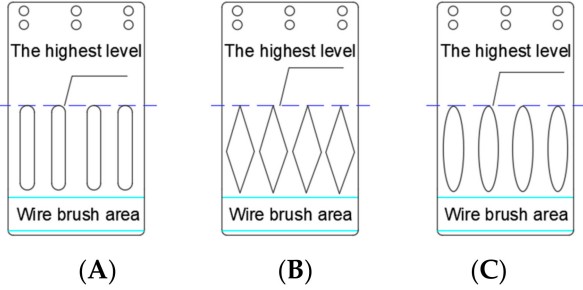

(A)　　　　　　　(B)　　　　　　　(C)

**Figure 10.** Three different brush plate groove shapes.

Taking the case where the brush plate entered the electrolyte at a depth of 200 mm as an example, the distribution value range of the dimensionless current density on the anode surface was calculated, and the results were plotted in Figure 11.

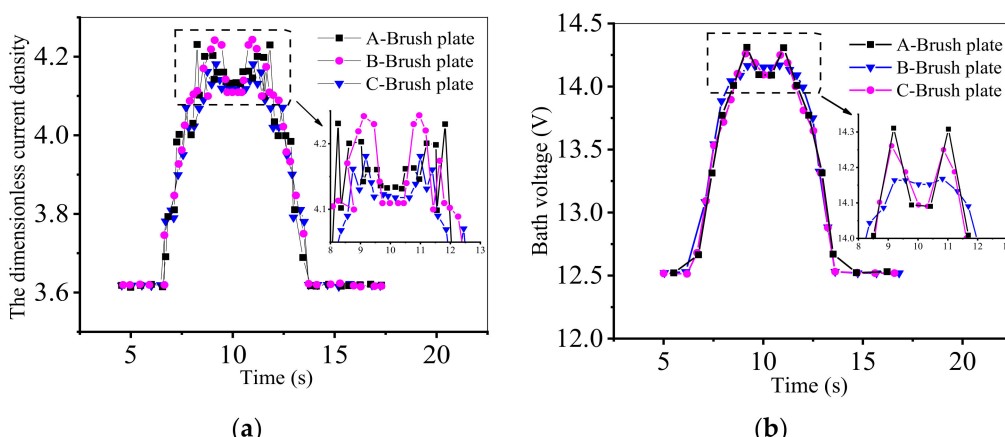

(a)　　　　　　　　　　　　　　　　(b)

**Figure 11.** *Cont*.

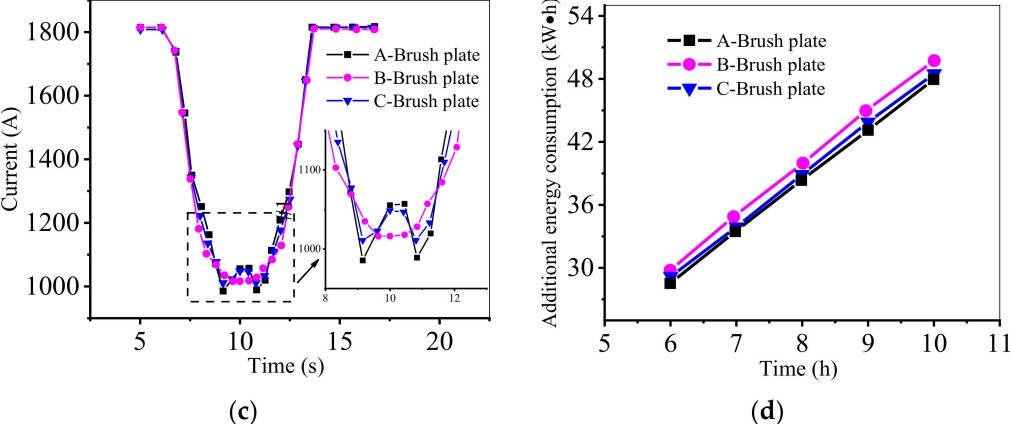

**Figure 11.** The variation of parameters with time. (**a**) The dimensionless current density; (**b**) Bath voltage; (**c**) Current; (**d**) Additional energy consumption.

It can be seen in Figure 11a that, given the same conditions of other parameters, the dimensionless current density distribution range of the elliptical slotting method was the smallest, followed by the circular slotting method and the largest was the diamond slot. In the nickel powder cleaning process, the change range of the current density distribution value of the C-brush plate was the smallest, and the current density distribution of the C-brush plate was more stable than that of the A-brush and B-brush plates.

Under the same electrolytic production conditions, the tank voltage changes caused by different brush plate structures also differed. When entering the electrolyte at a depth of 200 mm, the tank voltages of the A-, B-, and C-brush plates were 13.81, 13.99, and 13.87 V, respectively. The changes in the tank voltage and the current of the branch circuit where the brushed cathode plate was located during the nickel powder brushing process are shown in Figure 11b,c.

As seen in Figure 11b, the slot voltage fluctuations of the A-brush and C-brush plates were similar. However, the maximum slot voltages of the A-brush and the C-brush plates were 14.26 and 14.22 V, respectively, with a difference of 0.04 V. When the depth of the brush plate into the electrolyte was 400 mm, the voltage values of the two tanks were relatively close, with a difference of 0.01 V. When the B-brush plate approached the electrolyte at a depth of 400 mm, there was no apparent tank voltage fluctuation, and the change was relatively smooth.

It can be seen from Figure 11c that when the brush plate entered the electrolyte at a position of 400 mm, the current values under the A-brush and C-brush plates were relatively close at 1050.8 and 1045.9 A, respectively, with a difference of 4.9 A. However, the minimum current values under the A-brush and C-brush plates were 985.6 and 1006.9 A, respectively, with a difference of 21.3 A. The minimum current values of the C-brush and B-brush plates were close to 1006.7 and 1011.6 A, respectively. Meanwhile, changes of additional energy consumption with time for the three brush plates are shown in Figure 11d. The additional energy consumption under the A-brush plate was the lowest, followed by the C-brush plate in the middle, and then the highest for the B-brush plate.

Following the above analysis and the results of the dimensionless current density distribution on the anode surface, it can be stated that during the nickel powder cleaning process, the current density distribution is most uniform for the elliptical slotting method of those tested. The changes in the slot voltage and currents caused by the diamond-shaped slot brush plate were found to be relatively gentle. However, the diamond-shaped and circular groove brush plates had the highest and lowest energy consumption levels, respectively.

Because additional energy consumption is a vital brush plate indicator, it is reasonable to use the slotting method of the return groove when designing a brush plate to minimize this consumption.

### 4.1.2. Slot Area

To study the effect of different brush plate slot areas on the current field in the electrolyzer, other brush plate factors were controlled, the total grooving was changed, and three types of brush plates with different grooving area sizes were designed, as shown in Figure 12. The total area sizes of the grooves cut by a single brush plate were 60,000, 68,160, and 76,200 mm$^2$.

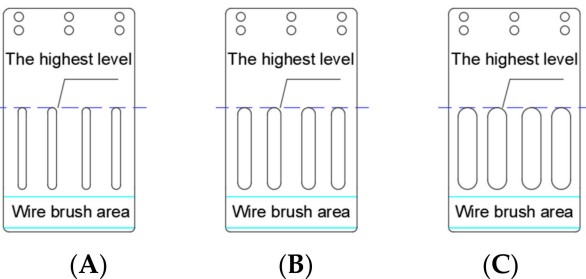

**Figure 12.** Three different slot area brush plates.

The changes in the dimensionless current density distribution on the anode surface during the nickel powder cleaning process are shown in Figure 13.

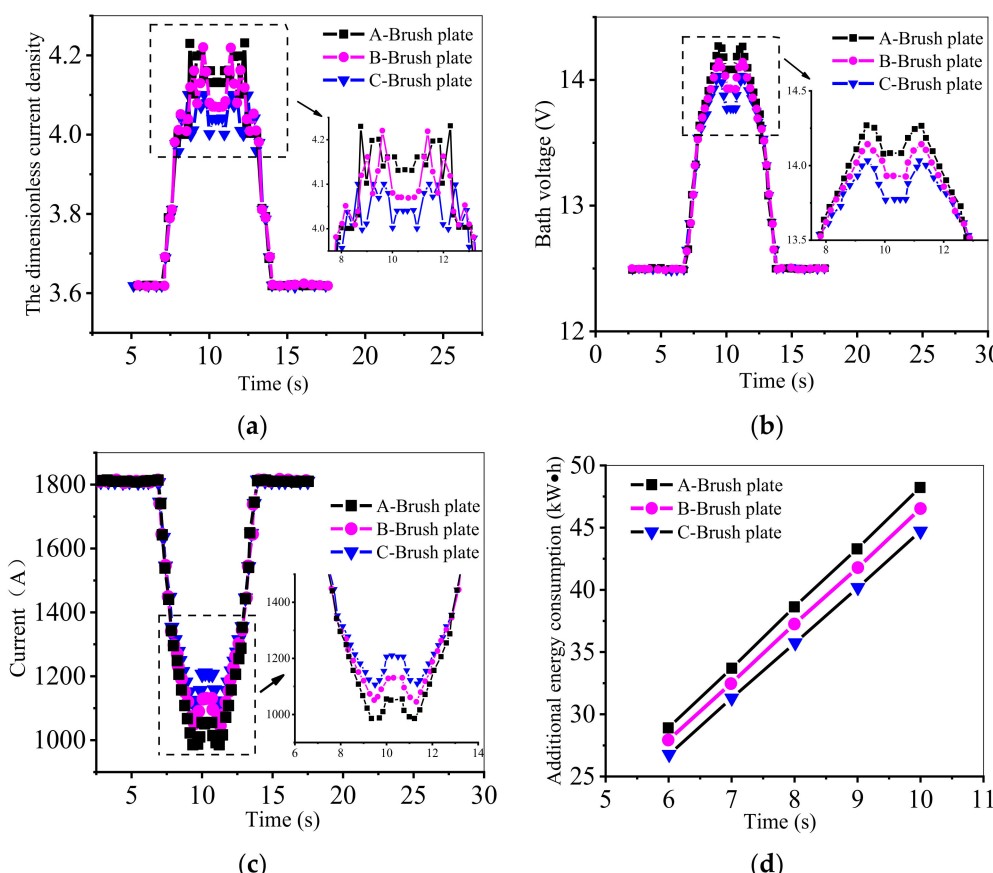

**Figure 13.** The variation of parameters with time. (**a**) The dimensionless current density; (**b**) Bath voltage; (**c**) Current; (**d**) Additional energy consumption.

As seen in Figure 13a, because the brush plate had the same brush filament area size and the slot area did not enter the electrolyte during these two periods of time, the dimensionless current density distribution values under brush plates with different slot areas were the same before t = 8 s and after t = 13 s. In the period of 8–13 s, the

nondimensional current density under the C-brush plate decreased and became more stable than the A-brush and B-brush plates.

The fluctuations of the voltages of the three brushing slots in the nickel powder cleaning process are shown in Figure 13b. The slot voltage of the three brush plates varied in the same pattern before 8 s and after 13 s, but the fluctuation of the slot voltage of the C-brush plate between 8 and 13 s was the smallest among the three brush plates.

The changes in current in the branch where the cathode plate was cleared according to the three brush plates are shown in Figure 13c. Among the three brush plates with different slot areas, the minimum current of the A-brush plate was 985.6 A, with a stable value of 1050.8 A; the minimum current of the B-brush plate was 1045.8 A, with a stable value of 1127.8 A; and the minimum current of the C-brush plate was 1103.5 A, with a stable value of 1205.8 A. The smallest amplitude of current fluctuation was caused by the C-brush plate, which had the largest slot area.

The changes of additional energy consumption with time for the three types of brush plate electrolyzers are shown in Figure 13d. The additional energy consumption of the electrolyzer decreased as the open tank area increased.

From the above analysis, the brush plates with large slot areas have less influence on the current field of the electrolyzer during nickel powder cleaning. At the same time, the additional energy consumption of the electrolyzer decreases as the open tank area increases.

### 4.1.3. Slot Position

In order to study the effect of the brush plate's slot position on the current field in the electrolytic tank, four types of brush plates were designed, as shown in Figure 14a–d. The other parameters of the slot were the same, but the relative position to the anode was different. The total area size of the grooves cut by a single brush plate was 60,100 mm². The position of the grooves on the A-brush and B-brush plates was consistent with the position of the center line of the anode, but the distribution method was different. Only one slot of the C-brush plate was opened in a position exactly between the two adjacent anodes, and all three slots of the D-brush plate were positioned between the two adjacent anodes.

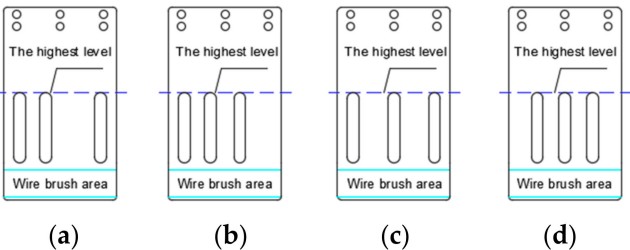

　　(**a**)　　　　　　(**b**)　　　　　　(**c**)　　　　　　(**d**)

**Figure 14.** Four different slot positions of the brush plate.

When the four kinds of brush plates entered 200 mm below the electrolyte surface, the distribution range of the dimensionless current density on the anode surface in the electrolyte changed with time, as shown in Figure 15.

It can be seen from Figure 15a that, in the period of 8–13 s, the range of the dimensionless current density distribution on the anode surface of the B-brush plate had a smaller variation range than that of the A-brush plate. A relatively concentrated slotting method resulted in a more uniform current density distribution on the anode surface in the electrolyte, provided that the slot position was the same as the relative position of the anode.

In the period of 8–13 s, the fluctuation of the dimensionless current density distribution value on the anode surface of the B-brush plate was smaller than that of the D-brush plate. The concentration of the slot arrangement on the B-brush and D-brush plates was the same, but the position of the slot on the B-brush plate was the same as the arrangement of the anode. The arrangement of the slot on the D-brush plate was the same as the arrangement of the gap between the adjacent anodes. The slotting method, with directly opposed anodes,

had less effect on the uniformity of the current density distribution on the anode surface in the electrolyte.

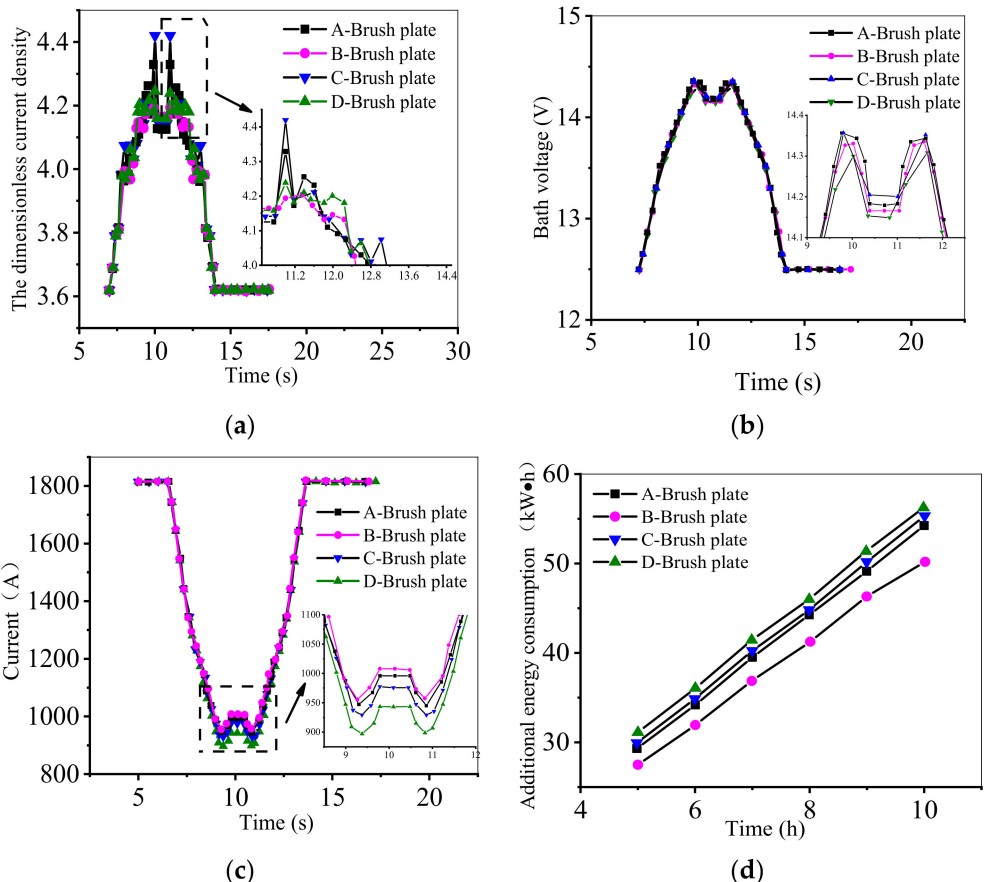

**Figure 15.** The variation of parameters with time. (**a**) The dimensionless current density; (**b**) Bath voltage; (**c**) Current; (**d**) Additional energy consumption.

In addition, the influence of the relative concentration of the slotting method was more evident than that of the relative position to the anode, as can be seen in the variation of the anode surface current density distribution values for the C-brush and D-brush plates.

The changes of electrolyzer tank voltage on the cathode plate during the nickel powder brushing process are shown in Figure 15b. The voltage variation under the C-brush plate was the largest, followed by the A-brush plate, then the B-brush plate, and finally the D-brush plate with the slightest variation. Overall, the voltage variations in the B-brush and D-brush plates were smaller than those of the A-brush and C-brush plates, indicating that the relatively concentrated and uniform grooving method had less effect on groove voltage.

As seen in Figure 15c, the minimum current of the A-brush plate was 950.4 A, with a stable value of 999.8 A; the minimum current of the B-brush plate was 956 A, with a stable value of 1005.9 A; the minimum current of the C-brush plate was 931 A, with a stable value of 977.3 A; and the minimum current of the D-brush plate was 902 A, with a stable value of 943.4 A. The magnitude of the current change in the A- and B-brush plates was close because the opening positions of the three slots on the two plates were the same as the relative positions of the anodes. The minimum and stable values of the current of the C-brush plate had small decreases relative to the A-brush and B-brush plates, while the D-brush plate had a more significant drop because in the C-brush plate, a slot was opened in the gap position between the two anodes, and the D-brush plate had three such slots. Using the slotting method, with anodes directly opposite to each other and relatively concentrated, it was possible to reduce the magnitude of current fluctuations. The changes in the energy consumption of the electrolyzer of the brush plates at different slot positions

over time are shown in Figure 15d. The slotting method, with the anodes arranged directly opposite each other and a relatively concentrated distribution, had the lowest additional energy consumption.

From the above analysis, it can be concluded that a relatively concentrated slotting method with directly opposed anodes can be used to make the current density distribution on the surface of an anode more uniform, decrease the slot voltage and current variation, and minimize additional energy consumption.

### 4.1.4. Number of Slots

In order to study the effect of the number of brush plate slots on the current field, three types of brush plates with different numbers of slots were designed, as shown in Figure 16. The total area of the groove cut in a single brush plate was 60,100 mm$^2$. Though one slot was opened at the center position of the A-brush plate, two slots of the B-brush plate were opened at a position just between two anodes, and three slots of the C-brush plate were all at a position between two adjacent anodes.

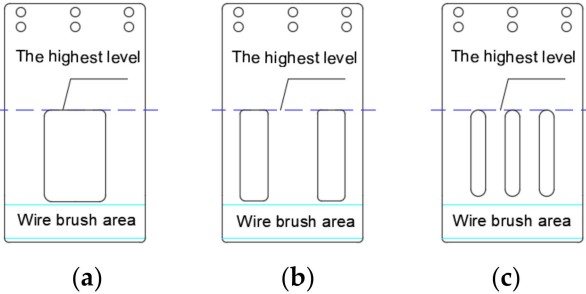

(a)   (b)   (c)

**Figure 16.** Three different types of brush plates with different numbers of grooves.

The dimensionless current density distributions on the anode surface when the three kinds of brush plates entered 200 mm below the electrolyte surface are shown in Figure 17.

As seen in Figure 17a, the magnitude of changes in the dimensionless current density distribution range on the anode surfaces of the A- and B-brush plates was relatively close over time, and the variation of the C-brush plate was smaller than that of the A- and B-brush plates. The dimensionless current density distribution of the C-brush plate and the current density distribution were more uniform and more stable than those of the other two brush plates.

In the process of cleaning the cathode plate with nickel powder, the variation law of the cell voltage of the brush plate electrolytic tank with different numbers of slots is shown in Figure 17b. The maximum value of the A-brush plate slot voltage was 14.47 V, with a stable value of 14.34 V; the maximum value of the B-brush plate slot voltage was 14.41 V, with a stable value of 14.27 V; and the maximum value of the C-brush plate slot voltage was 14.3 V, with a stable value of 14.16 V. It can be seen that the voltage increase in the C-brush plate slot with three slots under the same conditions was the smallest.

The changes in the current during the brush cleaning process with different numbers of slots are shown in Figure 17c. The minimum current value of the A-brush plate was 878.9 A, with a stable value of 924.2 A; the minimum current value of the B-brush plate was 894.9 A, with a stable value of 935.2 A; and the minimum current value of the C-brush plate was 902 A, with a stable value of 943.4 A. It can be seen that as the number of slots increased, the current reduction during the brush clearing process decreased.

The additional energy consumption of the electrolyzer with different numbers of slot brush plates varied with time, as shown in Figure 17d. As seen from the above graph, the additional electricity consumption decreased as the number of slots increased.

In summary, as the number of open slots increased and when other conditions were unchanged, the current density distribution on the anode surface in the electrolyte became

more uniform, there was less slot voltage and current variation in the branch circuit where the cathode plate was located, and the additional power consumption decreased.

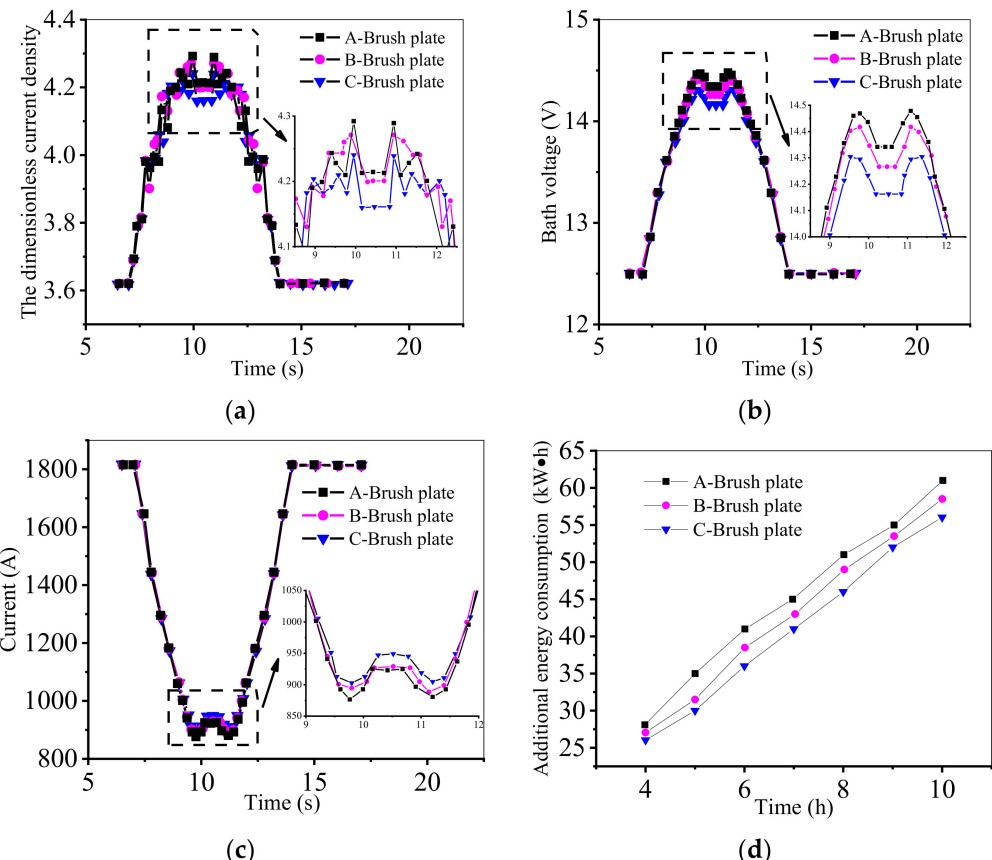

**Figure 17.** The variation of parameters with time. (**a**) The dimensionless current density; (**b**) Bath voltage; (**c**) Current; (**d**) Additional energy consumption.

## 4.2. Effect of Brush Plate Operating Parameters on Energy Consumption of the Electrolyzer

The main operating parameters of a brush cleaner are the brush cleaning speed and the pause time of the brush plate in the electrolyte. Consistent with the previous analysis, the influence of operating parameters of the existing C-brush plates on the voltage and current of the electrolytic cell was investigated. The C-brush plates as shown in Figure 18.

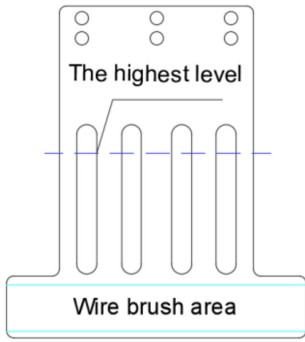

**Figure 18.** C-brush plate.

The changes in the brush plate slot voltage at different speeds in a cleaning cycle are shown in Figure 19.

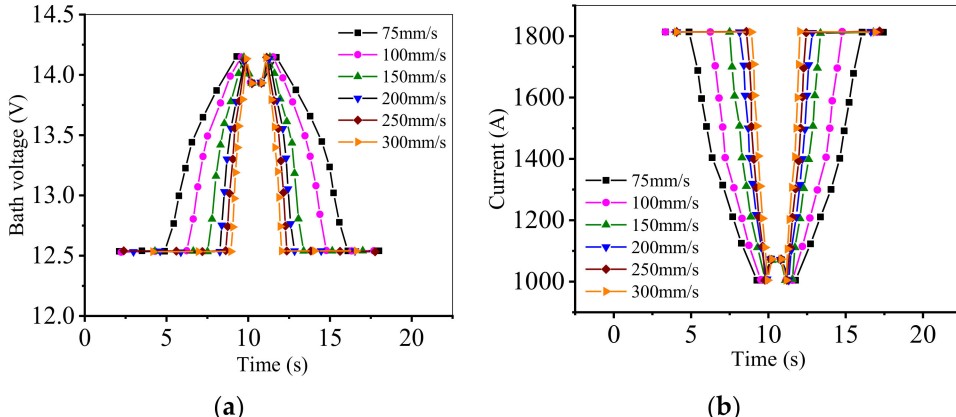

**Figure 19.** The variation of parameters with different brush cleaning speeds. (**a**) Bath voltage; (**b**) Current.

As seen in Figure 19a, during the operation of the brush cleaner, the fluctuation of the slot voltage caused by the brush plate at different brush cleaning speeds was about 1.6 V.

The change patterns of the current in a single cathode and anode set caused by the brush plate at different clearing speeds are shown in Figure 19b. The general trend of current change was essentially the same at different clearing brush speeds. When the cleaning speed was 75 mm/s, for example, the current was relatively stable from the moment the brush plate entered the electrolyte until 4.92 s. From 4.92 to 9.54 s, the current rapidly dropped until it reached a minimum value of 1005.1 A. The current increased from 9.54 to 10.2 s. At 10.2 s, the brush plate ran to 400 mm below the liquid level, and the current was 1073.8 A. The brush plate was suspended from 10.2 to 10.8 s, and the current was maintained at about 1073.8 A. After 10.8 s, the brush plate started to move up, and the curve of current change in the process of moving up and leaving the electrolyte was symmetrical with the curve of moving down. Throughout the operation of the brush plate, the maximum value of current fluctuation was 807 A, which was 44.5% of the normal current. It can be seen that the running speed of the brush plate had a significant effect on the current field in the electrolyzer.

The influence of the brush plate on the electrolytic cell over time varied with the cleaning speed, as shown in Figure 20.

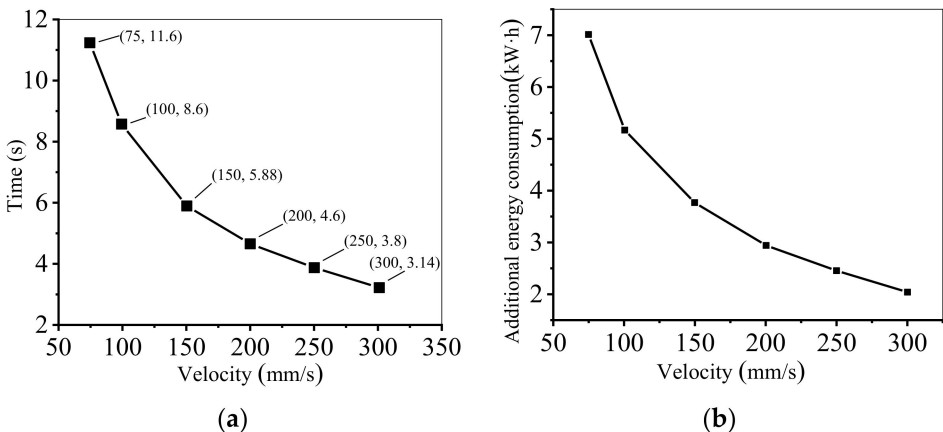

**Figure 20.** The variation of parameters with different brush cleaning speeds. (**a**) The influence time on the current field; (**b**) additional energy consumption.

Figure 20a shows that with the linear increase in the cleaning speed of the brush, the influence time of the brush in the electrolyte nonlinearly decreased, and the extent of

decrease also gradually diminished. It can be seen that when the clearing speed of the brush was increased to a certain extent, the speed increased by the same magnitude, and the influence time of the brush plate on the current field in the electrolyte only decreased by a small amount.

The variation pattern of the additional energy consumption with the clearing speed is shown in Figure 20b. The additional energy consumption continuously decreased as the speed increased, the same trend as that of the influence time of the cleaning speed on the electrolytic cell. Therefore, the impact time of a brush plate on a current field could be reduced by appropriately increasing the brush clearing speed and reducing the pause time of the brush plate in the electrolyte.

The impact time of the brush plate on the electrolyte consists of two parts: the brush cleaning time and the pause time of the brush plate in the electrolyte, though the pause time of the brush plate in the electrolyte is the same at different cleaning speeds. From the analysis results, it can be seen that when the brush plate speed exceeded 200 mm/s, the influence time of the brush plate on the current and voltage in the electrolyte was slightly reduced, and because the brush plate speed was too fast, it affected the safety and stability of the system. For that reason, the brush plate speed can be set to a maximum of 200 mm/s. Considering that the minimum unit of the action time of the system brush plate commutation was 0.2 s, the pause time of a brush plate in an electrolyte can also be adjusted to 0.2 s.

## 5. Brush Plate Optimization Design

There is little research on the structure of brush plates in the industry. Based on the analysis of the structural parameters of the brush plate, the design of the brush plate structure can be optimized, the D-brush plate was proposed. A comparison with the existing C-brush plate is shown in Figure 21.

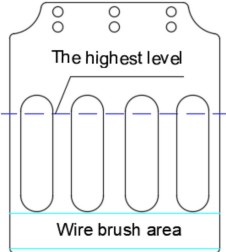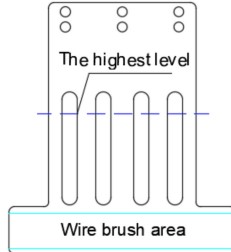

**Figure 21.** Comparison of D-brush and C-brush plates.

Figure 21 shows that the D-brush plate has a similar slot shape to the C-brush plate. In terms of slot area, the return slot area of the D-brush plate is larger than that of the C-brush plate, and the width of the return slot is close to the width of the anode in the electrolyzer. To reduce the negative impact on the structural stiffness of the brush plate due to the length of the return groove, the length of the original C-brush plate above the liquid level of the return groove is reduced in the D-brush plate. In the slot position, the D-brush plate adopts a slotting method that faces the anode and is relatively concentrated. Moreover, the lower position of the return slot is lowered to mitigate the influence of the slot-empty area on the current field.

## 6. Analysis of the Impact of the Optimized Brush Plate on the Energy Consumption of the Electrolyzer

The D-brush plate was modeled in the same way as the model built in the previous section. Following the previous analysis, the brushing speed of the D-brush plate was set to 200 mm/s, and the pause time was set to 0.2 s. The other production processes were the same. The influence pattern of the D-brush plate on the electrolytic cell was studied, and the results were compared with the influence situation of the existing C-brush plate. The

dimensionless current density distributions of D-brush and C-brush plates on the anode surface in the electrolytic, with the plates entering the electrolyte at a depth of 200 mm as an example, are shown in Figure 22.

As shown in Figure 22a, the variation amplitude of the dimensionless current density distribution of the C-brush plate was 0.54, while the variation amplitude of the D-brush plate was 0.35, a 35% reduction. In addition, it took 7 s for the C-brush plate to complete the cleaning process of one cathode plate, while for the D-brush plate, after optimizing the operation parameters, it only took 4.2 s, and the impact time on the current field of the electrolytic cell was reduced by 40%.

The changes in the tank voltage caused by the D-brush and C-brush plates during the nickel powder cleaning process are shown in Figure 22b. It can be seen that in the nickel powder clearing process, the increase in the amplitude of the tank voltage of the C-brush plate was 1.63 V, while that of the D-brush plate was 1.07 V. The tank voltage was 1.41 V higher than normal when the C-brush plate was suspended in the electrolyte. In contrast, that for the D-brush plate was 0.67 V. The increase in the amplitude of the tank voltage of the D-brush plate was reduced by 34% compared with that of the C-brush plate. The increase in the amplitude of the tank voltage was reduced by 52% during the pause in the electrolyte.

When the D-brush and C-brush plates were cleaned, the current changes in the branch where the cathode plate was located are shown in Figure 22c. In the nickel powder cleaning process, the current reduction amplitude of the C-brush plate was 806.9 A, while that of the D-brush plate was 491.7 A. When the brush plate was suspended in the electrolyte, the current of the C-brush plate was 738.2 A, lower than the normal value, while that of the D-brush plate was 323.2 A. It can be seen that the current drop amplitude of the D-brush plate was reduced by 39% compared to that of the C-brush plate, and the current drop amplitude of the D-brush plate was reduced by 56% when suspended in the electrolyte.

The variations of the additional energy consumption of the D-brush and C-brush plates with time are shown in Figure 22d, the additional energy consumption per unit time of the D-brush plate was reduced by 50.9% compared to that of the C-brush plate, and the effect of energy saving was noticeable.

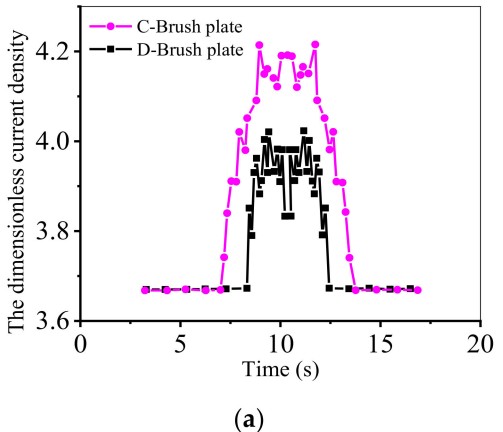

(a)

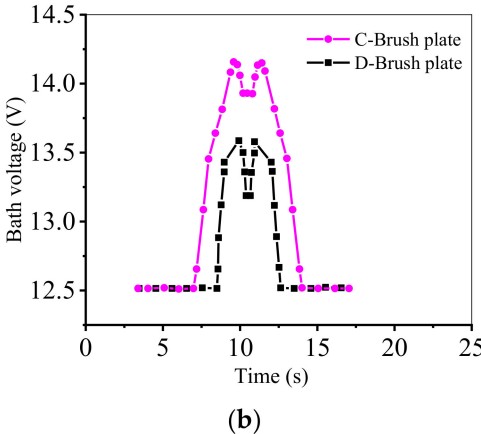

(b)

**Figure 22.** *Cont.*

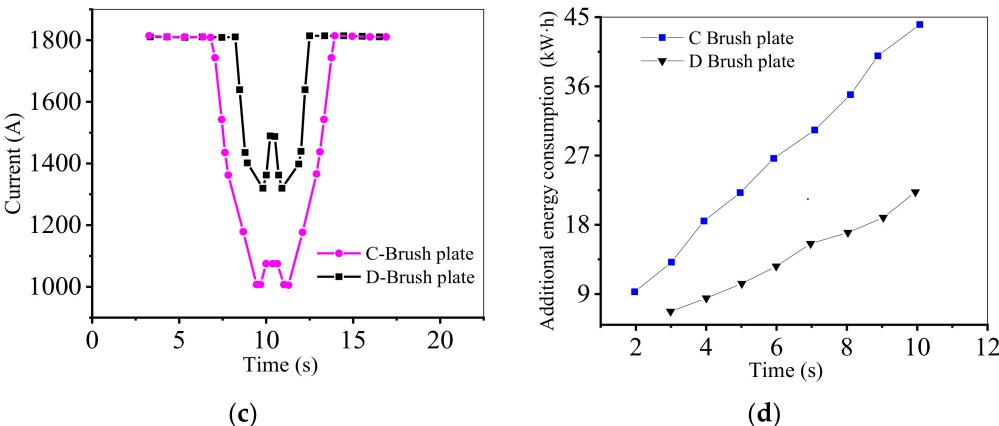

**Figure 22.** The variation of parameters with time. (**a**) The dimensionless current density; (**b**) Bath voltage; (**c**) Current, (**d**) Additional energy consumption.

## 7. Conclusions, Challenges, and Further Work

### 7.1. Conclusions

In this paper, the electrolyte is chloride, the brush plate of a cleaning machine was taken as the research object, and the effect of the energy consumption of the brush plate in the electrolytic cell was analyzed. The main results in this paper are summarized as follows:

(1) The influence of the structural and operating parameters of the brush plate on the current density distribution on the anode surface, as well as the cell voltage and current changes, was analyzed. According to the results, the structure of the commonly used empirical C-brush plate was optimized, and the optimized D-brush plate was proposed. We optimized the clearing speed of the brush plate to 200 mm/s and the pause time in the electrolyte to 0.2 s, which not only met the safety and stability requirements of the equipment but also allowed the brush to have a small effect on the current field in the electrolyte.

(2) Considering various aspects, the D-brush plate was superior to the C-brush plate. In the nickel powder cleaning process, the change amplitude of the dimensionless current density distribution on the anode surface under the D-brush plate decreased by 35%, and the tank voltage decreased by 34%. The drop amplitude of the branch current where the cathode plate was cleaned was reduced by 39%. The impact time on the current field of the electrolyzer was reduced by 40%, and the additional energy consumption was reduced by 50.9%. The optimized brush plate will not have an adverse effect on the environment. The research result can not only help nickel powder enterprises in the production process to reduce energy consumption and develop a circular economy but also provide a new theoretical reference for the structural design of nickel powder brush plates.

### 7.2. Challenges and Further Work

At present, there are induced electric field and induced magnetic field in the electrolytic cell. Although they have little effect on the electrolyzer, they still have an impact on the current field between the cathode and the anode. Lack of understanding of their specific effects raises a challenge for in-depth research. Nevertheless, we intend to conduct analytical studies through a large number of experiments in the future and will also consider placing the newly proposed D-brush plate into different electrolytes for analysis, such as sulfate electrolytes.

**Author Contributions:** Conceptualization, S.Y., Z.Y. and L.Z.; methodology, S.Y. and Z.Y.; validation, S.Y., Z.Y. and L.Z.; formal analysis, S.Y. and Z.Y.; investigation, S.Y., L.Z. and G.Z.; writing—original draft preparation, S.Y.; writing—review and editing, S.Y., Z.Y. and L.Z.; funding acquisition, Z.Y. All authors have read and agreed to the published version of the manuscript.

**Funding:** This work was funded by the project of intelligent new energy loader of China (XQ201828).

**Institutional Review Board Statement:** Not applicable.

**Informed Consent Statement:** Not applicable.

**Data Availability Statement:** Not applicable.

**Conflicts of Interest:** The authors declare no conflict of interest.

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
