# Peer review of "The Effect of Brush Plate Structure and Operating Parameters on the Energy Consumption of Electrolytic Cells"

_processes, doi:10.3390/pr9122186_

Round 1

Reviewer 1 Report

Manuscript ID: processes-1468809

Manuscript Title: The Effect of Brush Plate Structure and Operating Parameters on the Energy Consumption of Electrolytic Cells

Reviewer Comments:

General comment:

This paper is regarding a research on effect of brush plate structure and operating parameters on the energy consumption of electrolytic cells. However, the writing and presentation of data is not of publication quality. The script in this current form can be revised to achieve publication quality. There are some clarifications needed to understand the processes carried out in this work. To conclude, this paper needs to be revised carefully before it can be considered in journal like Processes. Hope that comments below will be able to help to further improve the paper

  • Please make sure that the paper is checked by native English speaker, the language needs improvement.
  • Please check Guides for Authors to make sure it is followed strictly
  • Language: There are some language errors (tenses, singular/plural) and incomplete sentences in the script. Please check the sentence structure, tenses and language carefully in the revised manuscript. There are some spelling mistakes occurred even in the Introduction itself.
  • Take note of unit spacing issue and citation spacing issue
  • Be consistent with the unit notation, whether s/mm or mm-1

Abstract:

  • Needs minor revisions prior to the amendment of the main content.
  • An abstract is often presented separately from the article, so it must be able to stand alone. Hence the problem statement, aim, novelty and results of the study have to be included in.
  • The abstract is rather long, maybe can consider shortening it and highlight the main points

Introduction:

  • Describe more on the environmental and performance issues together with treatment possibility and methods
  • Highlight novelty and the focus of this study in last paragraph.
  • Kindly refer papers below as they are highly relevant to this report:
  • “A review on conventional and novel materials towards heavy metal adsorption in wastewater treatment application” Journal of Cleaner Production
  • “Natural hydroxyapatite from fishbone waste for the rapid adsorption of heavy metals of aqueous effluent” Environmental Technology & Innovation

Main body:

  • The main objective and novelty of this work is still deemed not highlighted enough. The authors should put in more efforts to revise the discussion properly in order to let the authors understand the importance of this work.
  • Kindly improve on the discussion and critically evaluate what is the significance of the results of the work. Include more relevant literatures.
  • Many large sentences, kindly break them for better understanding
  • The overall structure needs to be improved
  • Standardize the symbols and units
  • Try to combine some figures together in the analysis section

Conclusion

  • Kindly improve to include in more concise and significant results.
  • Should include some present challenges and possible routes to improve them. Describe them in more details.

Figures and Tables

  • Is it possible to integrate some of the figures together as the whole MS consists of many figures and might be difficult for readers to relate later on?

Papers suggested for reading:

  • “Effects of foam nickel supplementation on anaerobic digestion: direct interspecies electron transfer” Journal of Hazardous Materials

Reviewer 2 Report

1)What is the effect of electrode structure on low concentration electrodeposition, how can the simulation structure guide electrode structure design?

2)The main energy consumption source of electrolytic cell? Anode, cathode?

3)Is there not enough analysis or discussion in the introduction? Include appropriate analysis in the introduction, energy generation analysis?

4)What specific guidance do simulations and calculations have for the structural design of electrodes and brushes? Magnification, shape, structure?
